# Recommending Cryptocurrency Trading Points with Deep Reinforcement Learning Approach

**Otabek Sattarov [1], Azamjon Muminov [1], Cheol Won Lee [1], Hyun Kyu Kang [1], Ryumduck Oh [2], Junho Ahn [2], Hyung Jun Oh [3] and Heung Seok Jeon [1],***

[1]  Department of Software Technology, Konkuk University, Chungju 27478, Korea;
    otabeksattarov95@gmail.com (O.S.); azammuminov92@gmail.com (A.M.); e10000won@gmail.com (C.W.L.);
    hkkang@kku.ac.kr (H.K.K.)
[2]  Department of Software and IT convergence, Korea National University of Transportation,
    Chungju 27469, Korea; rdoh@ut.ac.kr (R.O.); jhahn@ut.ac.kr (J.A.)
[3]  Department of Computer Information, Yeungnam University College, Gyeongsan 38541, Korea;
    ohlake@ync.ac.kr
*  Correspondence: hsjeon@kku.ac.kr; Tel.: +82-43-840-3621

**Abstract:** The net profit of investors can rapidly increase if they correctly decide to take one of these three actions: buying, selling, or holding the stocks. The right action is related to massive stock market measurements. Therefore, defining the right action requires specific knowledge from investors. The economy scientists, following their research, have suggested several strategies and indicating factors that serve to find the best option for trading in a stock market. However, several investors' capital decreased when they tried to trade the basis of the recommendation of these strategies. That means the stock market needs more satisfactory research, which can give more guarantee of success for investors. To address this challenge, we tried to apply one of the machine learning algorithms, which is called deep reinforcement learning (DRL) on the stock market. As a result, we developed an application that observes historical price movements and takes action on real-time prices. We tested our proposal algorithm with three—Bitcoin (BTC), Litecoin (LTC), and Ethereum (ETH)—crypto coins' historical data. The experiment on Bitcoin via DRL application shows that the investor got 14.4% net profits within one month. Similarly, tests on Litecoin and Ethereum also finished with 74% and 41% profit, respectively.

**Keywords:** trading; machine learning; deep reinforcement learning; moving average; double cross strategy; day trading; swing trading; position trading; scalping

## 1. Introduction

Over half a century, a significant amount of research has been done on trading volume and its relationship with good point returns [1–12]. Although the existence of a relationship between trading stock prices and future prices is incompatible with a weak form of market efficiency [10], the exploration of the relationship has received growing attention from researchers and investors. One of the reasons for the great attention to these relationships is that many believe that price movements can bring sufficient income if it is right to decide on the volume of trading.

Traders' experience shows that catching up the good points for trading is not easy. Each trader or financial specialist realizes that during the trade procedure, one of three activities happens: now and again, the trader buys coins from the market; once in a while, the trader will sell or hold up until the best minute comes. The ultimate goal is to optimize some relevant performance indicators of the trading system, such as profit by implementing whatever theorem or equation. Over the past

few decades, scientists in the economic fields have studied the coin market changes and factors that affect the market. At the end of their research, they invented techniques and strategies which can suggest more effective actions and help catch good trade points. Below, we take a quick tour through five of the most common active trading strategies used by investors in financial markets. In addition, they are commonly used between traders and differed from other strategies with their simpleness to understand.

1.  The Double Crossover Strategy. This strategy uses two different price movement averages: long-term and short-term. By their crosses defines the golden cross and death cross, which indicates whether long-term bull market: going forward price, or long-term bear market: going downward price. Both of them relate to firm confirmation of a long-term trend by the appearance of a short-term moving average crossing the main long-term moving average [13].
2.  Day trading, as its name implies, is speculation in securities, in particular, the purchase and sale of financial instruments during one trading day, so that all positions are closed before the market closes during the trading day. Day traders exit positions before the market closes to avoid unmanageable risks and negative price gaps between one day's close and the next day's price at the open [14].
3.  Swing trading is a speculative trading strategy in the financial markets where the traded asset is held for one to several days in order to profit from price changes or "swings" [15]. Profits can be made by buying an asset or selling short sales.
4.  Scalping is the shortest period of time in trading in which small changes in currency prices are used [16]. Scalpers create a spread, which means buying at the Bid price and selling at the Ask price to get the difference between supply and demand. This procedure allows making a profit even when orders and sales are not shifted at all if there are traders who are willing to take market prices [17].
5.  Position trading involves keeping a position open for a long period of time. As a result, a position trader is less concerned with short-term market fluctuations and usually holds a position for weeks, months, or years [18].

During our research, we experienced to trade several types of cryptocurrencies in some periods based on the above-mentioned strategies. Unfortunately, some results are not pleasingly. We obtained that while strategies worked well, and the trading process finished with a massive amount of benefit. However, in some cases, we lose money instead of earning. The results of the experiment will be shown in the fourth section in detail. One of the reasons for losing money is as follows. The strategies keep bringing profit when all participants in the trade market keep their temporary position. In real life this case happens rarely. Sometimes traders want to sell owned coins, although strategy requires to purchase, or trader cannot find an available coin for purchase from the market. Alternatively, some of them believe the theorem called "Random walk theory" and do not care about any rules. This theory was popularized in 1973 when author Burton Malkiel wrote a book called "A Random Walk Down Wall Street" [19]. The theory suggests that changes in stock prices have the same distributions and are independent on each other. Accordingly, the past changes in price or trend of a stock price cannot be used to predict its future movement. Shortly, stocks take a random path and are impossible to predict.

Simultaneously, the researchers in the field of computer science have studied the stock market and tested their modules and systems. There have been previous attempts to use machine learning to predict fluctuations in the price of bitcoin. Colianni et al. [20] reported 90% accuracy in predicting price fluctuations using similar supervised learning algorithms; however, their data was labeled using an online text sentiment application programming interface (API). Therefore, their accuracy measurement corresponded to how well their model matched the online text sentiment API, not the accuracy in terms of predicting price fluctuations. Similarly, Stenqvist and Lonno [21] utilized deep learning algorithms, on a much higher frequency time scale of every 30 min, to achieve a 79% accuracy in predicting bitcoin price fluctuations using 2.27 million tweets. Neither of these strategies used data labeled directly

based on the price fluctuations, nor did they analyze the average size of the price percent increases and percent decreases their models were predicting. More classical approaches of using historical price data of cryptocurrencies to make predictions have also been tried before. Hegazy and Mumford [22] achieved 57% accuracy in predicting the actual price using a supervised learning strategy. Jiang and Liang [23] utilized deep reinforcement learning to manage a bitcoin portfolio that made predictions on price. They achieved a 10× gain in portfolio value. Last, Shah and Zhang [24] utilized Bayesian regression to double their investment over 60 days. Fischer et al. [25] have successfully transferred an advanced machine-learning-based statistical arbitrage approach. More relevant research to our paper has been done by A.R. Azhikodan et al. [26] aimed to prove that reinforcement learning is capable of learning the tricks of stock trading.

Similarly, John Moody and M. Saffell [27] used reinforcement learning for trading, and they compared trading results achieved by the Q-Learning approach with a new Recurrent Reinforcement Learning algorithm. The scientist Huang Chien-Yi et al. [28] proposed a Markov Decision Process model for signal-based trading strategies that give empirical results for 12 currency pairs and achieve positive results under most simulation settings. Additionally, P.G. Nechchi et al. [29] presented an application of some state-of-the-art learning algorithms to the classical financial problem of determining a profitable long-short trading strategy.

However, as a logical continuation of previous attends to use machine learning for trading, we would like to propose a deep reinforcement learning approach that learns to act properly in the stock market during trade and serves to maximize trader's profit. Generally, an application works as follows. To start, it takes a random action $a_1$ (for example, it buys) and moves to the next action $a_2$ (sells). After selling, the reward is estimated by subtracting the selling price with the buying price $r = a_2 - a_1$. If the result is positive $r > 0$, an agent will get a positive reward. If $r < 0$, the agent will be punished with negative reward. Based on the value of the reward, the application realizes the quality of its actions and uses it to improve the skill of how to take the right action.

This paper proposes an application that recommends taking the right action and maximize investor's income by using a deep reinforcement learning algorithm.

The paper comprises of 5 sections. Section 1 is a presentation that gives brief information about the trade procedure, systems, and related researches to our topic, Section 2 is a background of the paper that enables us to see how essential to utilize a deep reinforcement learning approach; Section 3 hypothetically portrays the proposed application; the Section 4 examines the analysis procedure and results; lastly, Section 5 finishes up our paper.

## 2. Background

In the introduction part, we briefly introduced five common active trading strategies. In this section, we will try to dive deeper into them. For an explanation of how they are useful in real life, we downloaded bitcoin hourly historical price data and tested them with some of these strategies. Indeed, any kind of historical data (minutely, hourly, daily, and weekly) are available on the web sources, and most of them are free to use. The next step of the experiment is trading in the cryptocurrency market by strategy rules' guidance.

### 2.1. The Double Crossover Strategy

As noted above, there are two kinds of price averages required for this strategy. The longer calculating period (in our case 60 days price average) is called the long-term moving average (LMA) and the shorter period called the short-term moving average (SMA—15 days price average). To calculate the simple moving average, whether it is ten days or 60 days, one adds the price of 10 days and divides into ten or sums up the 60 days and divides into 60. For the next day calculation, drop the last data and add the latest data point and recalculate the average. In this way, the average keeps on

moving forward with every new data point being generated. Below, Equation (1) is a quick explanation of how to calculate ten days moving average, where *d*—daily price, *p*—points, *n*—number of points:

$$p_1 = \frac{d_1 + d_2 + \cdots + d_9 + d_{10}}{10}; p_2 = \frac{d_2 + d_3 + \cdots + d_{10} + d_{11}}{10}; \ldots; p_n = \frac{d_{n-9} + d_{n-8} + \cdots + d_{n-1} + d_n}{10} \qquad (1)$$

According to the strategy, next, we should find the right crossover points. To find crossovers, we can use a rule that explained in the strategy:

1. Downtrend changes to the uptrend, the price crosses over the short-term, and secondly, the short-term crosses over the long-term. This is called the "Golden Cross".
2. Uptrend changes to the downtrend, firstly, the price crosses under the short-term, and secondly, the short-term crosses under the long-term. This is called "Death Cross".

The golden cross interpreted as a signal to a final upward turn in a market by analysts and traders. So, the operation tactics can be "buy at the golden cross". Conversely, a similar intersection of the lower moving average is a deadly cross, and it is believed that it signals a decisive decline in the market. So, it is a useful act like "sell at death cross".

We downloaded hourly historical data of Bitcoin from 2 of October 2016 to 1 of the March 2019 year and traded based on the Double Cross strategy. The trade process visually described in Figure 1. Trade process went as follows: for starting, the trader's capital was 10 000 $ cash and 20 Bitcoin (BTC) coin, and he/she acted with the guidance of strategy: bought at the golden cross, sold at the death cross.

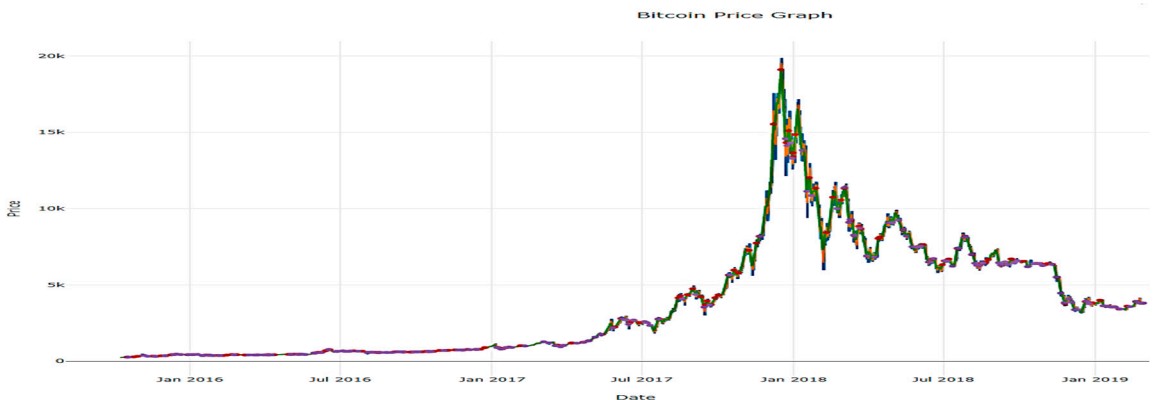

**Figure 1.** The Bitcoin hourly data diagram from the 2 of October 2016 to the 1 of March 2019. The Dead and Golden Crosspoints matched with green and red points, respectively.

One factor that every investor must pay attention to is transaction costs. They are important because it is one of the key determinants of net returns. High transaction costs can mean thousands of dollars lost from not just the costs themselves, but because the costs reduce the amount of capital available to invest [30].

Table 1 presents the main information about the process: invested money for trade includes sum of cash and value of available coins at that moment, number of appeared golden and death Crosspoints, sum of trading fee that is 0.15% of the trading value taken whenever trader acts, money after trading also includes sum of all cash and available coins value at the final trade process, quality of trade process, whether the capital is lost or gained profits. For evaluating the trading process, we show two cases in Table 1. The first one shows how invested money changes when the trader's act goes based on the Double Crossover strategy's guidance. The second case describes when a trader holds invested money and coins until the final moment of trading without performing any actions, then compares the initial investment with the final capital.

**Table 1.** Main information about the trading process.

| Invested Money for Trading ($) | Trade with | Number of "Buy" Actions | Number of "Sell" Actions | Transaction Fee ($) | Money after Trading ($) | Quality of Trading (%) |
|---|---|---|---|---|---|---|
| 22 223.4 | DC strategy | 121 | 111 | 1 822.3 | 73 779.1 | Grew 332.0 |
| | hold position | 0 | 0 | 0 | 86 248 | Grew 388.0 |

According to Table 1, the quality of the trade process when applied Double Crossover strategy is 56% lower than trading only with the "hold" action. These results indicate one of the weaknesses of the strategy. However, the single result is not enough to judge the whole strategy. For that reason, we will test the strategy differently.

Continuously, to test the strategy more deeply, we divided our last prepared data into two parts. The first part involves the data where cryptocurrency price is wildly raising, and the second part includes the data where the price is decreasing. The Double Crossover strategy was applied to both parts. Table 2 is shown the main trading criteria such as prepared money for trading, the number of appeared Golden and Death Crosspoints, all money spent for the transaction, money after the trade process, and the column for the process evaluation.

**Table 2.** Main information about the trading process for the first part (increasing period).

| Data (Part) | Trade with | Invested Money for Trading ($) | "Buy" Actions | "Sell" Actions | Transaction Fee ($) | Money after Trading ($) | Quality of Trading (%) |
|---|---|---|---|---|---|---|---|
| First | DC strategy | 22 223.4 | 54 | 40 | 341.1 | 450 529.2 | Grew 2027.3 |
| | hold position | | 0 | 0 | 0 | 399 500 | Grew 1797.6 |
| Second | DC strategy | 396 500 | 65 | 70 | 1441.9 | 105 611.5 | Lost 73.4 |
| | hold position | | 0 | 0 | 0 | 86 486 | Lost 78.2 |

According to Table 2 information, during the experiment on the first part of data, there are 54 Golden and 40 Death Crosspoint occurred. As a result of the strategy, the trader gets 20.2 times more profit, which is very well. Additionally, trade with the Double Crossover strategy brings a noticeable 228% more profit than trade only with "hold" action. However, the experiment with the second part of data shows 70 "sell" actions and 65 "buy" actions, and in consequence trader's money decreased by 73.4%. Unfortunately, for this period, the strategy did not give expected results: instead of profiting, the trader lost more than 73% of the invested money, and it is 5% higher than if he/she does not use the strategy.

The above-given experiment shortly tells the strategy can be useful as it can be dangerous for the trader's capital.

*2.2. Day Trading*

Day trading is perhaps the most famous style of active trading. This is often considered the pseudonym of the most active trading. Day trading is the method of buying and selling securities within the same day. Positions are closed on the same day they were occupied, and no position is held during the night. Traditionally, day trading is carried out by professional traders, such as specialists or market makers [31].

Due to the nature of financial leverage and possible quick returns, daily trading results can vary from extremely profitable to extremely unprofitable, and high-risk traders can generate either huge interest incomes or huge interest losses [32]. Below are a few basic trading methods by which day traders try to make a profit.

- News playing. The basic strategy of playing a game in the news is to buy a stock that has just announced good news, or a short sale on bad news [33].

- Rebate trading. Rebate trading is a stock trading style in which electronic communication networks (ECN) rebates are used as the main source of profit and income [34].
- Contrarian investing. Contrarian investing is a market timing strategy used in all trading periods [35].
- Day traders are typically well-educated and well-funded. For day trading, there is no such kind of specific rule that indicates to the buy or sell. The day traders are looking for any good or bad market news. That is the reason, no need to test historical price data with day trading strategy. In addition, researchers have discovered 24 very surprising statistics [36] about day trading such as:
- 80% of all day traders leave within the first two years.
- Among all day traders, almost 40% of traders trade only one month. Within three years, only 13% continue day trading. Five years later, only 7% remained.

Day traders with good past performance continue to make good profits in the future, although only about 1% of day traders can predictably make a profit minus fees.

These statistics tell that day trading strategy might be dangerous for non-specialist traders.

### 2.3. Swing Trading

Swing trading is a trading methodology that seeks to capture a swing (or "one move"). Swing trading has been described as a form of fundamental trading in which positions hold longer than one day. Most fundamentalists are swing traders since it usually takes several days or even a week to change the fundamental indicators of a company in order to cause a sufficient price movement to make a reasonable profit. Figure 2 graphically shows the basic terms of swing trading.

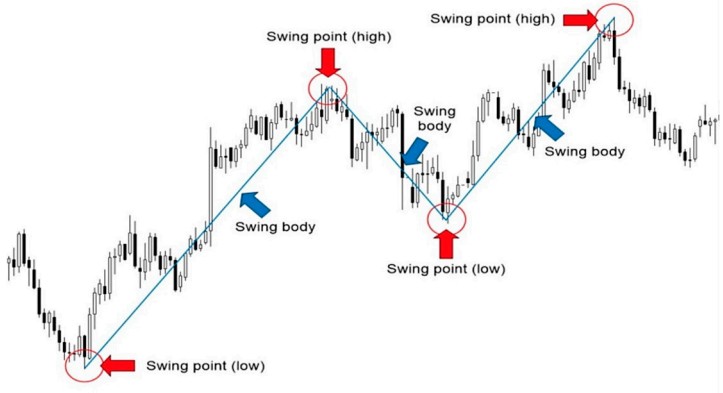

**Figure 2.** The basic term of Swing trading.

In reality, Swing trading sits in the middle of the continuum between day trading to trend trading. A day trader will hold a stock anywhere from a few seconds to a few hours but never more than a day; a trend trader examines the long-term fundamental trends of a stock or index and may hold the stock for a few weeks or months. Swing traders hold a particular stock for a while, generally a few days to two or three weeks, which is between those extremes, and they will trade the stock based on its intra-week or intra-month oscillations between optimism and pessimism [37].

A set of mathematically based objective rules can be used to create a trading algorithm using technical analysis to give buy or sell signals. Simpler rule-based trading strategies include the technique of Alexander Elder, which tests an instrument's market trend activity using three separate, moving average closing prices. Only when the three averages are moving in an upward direction is the instrument traded Long and only traded Short when the three averages shift downward [38]. The experiment and its result will be discussed in Section 4.

Swing trading tips, like all investment strategies, are never to lock yourself into a specific set of rules but rather guidelines. If something is evident, traders do not force themselves to ride it out just

because the strategy dictates. Swing trading is a middle of the road investment strategy and should be considered when developing a personal approach.

## 2.4. Scalping Trading

Scalping is probably the fastest procedure utilized by dynamic merchants. It incorporates using different value holes about by bid-ask spreads and request streams. The methodology by and large works by making the spread or purchasing at the bid price and selling at the asking price that value gets the distinction between the two value focuses. Bid-ask spread is described in Figure 3. Scalpers endeavor to hold their situations for a brief period, in this way diminishing the hazard related to the technique.

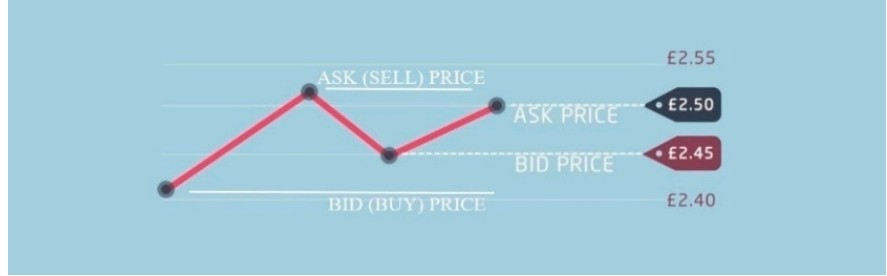

**Figure 3.** Make a profit between the bid and ask price difference.

Additionally, a scalper does not attempt to exploit massive moves or move high volumes. Instead, they struggle to require advantage of little moves that often occur and move smaller volumes additional typically. Since the amount of profits per trade is tiny, scalpers explore for other liquid markets to extend the frequency of their trades. In contrast to swing traders, scalpers like quiet markets that are not at risk of fast worth movements so that they will probably build the repeatedly unfold on identical bid/ask costs. Scalpers have different methodologies that can tell whether buying or selling. Mostly, they are using three different moving averages short-term charts. One of them called Moving Average Ribbon Entry Strategy, which is used 5-8-13 simple moving average combination on the two-minute chart to identify strong trends that can be bought or sold.

However, economists suggest not to believe positive things about scalping. There is no single verified formula that guarantees you scalping success in at least 90% of cases. Similarly, if something sounds too good to be true, it is most likely true, especially in an atmosphere of forex scalping.

As always, it is true that the investment will be under risk, so forex traders can benefit from doing their due diligence and/or advising independent financial advisors before engaging in ranges trading or other strategies.

## 2.5. Position Trading

A trader who holds a position in an asset for a long period of time is called a position trader. The period can vary from several weeks to years. In addition to "buy and hold", this is the longest retention period among all trading styles. Positional trading is largely the opposite of day trading. The position trader is generally less concerned about the short-term driving force of asset prices and market corrections, which may temporarily change the price trend. For this reason, position traders often make a nominal number of transactions during the year, adhering only to critical movements, and not to the "chop-and-change" approach. They make hundreds of trades, while day traders make thousands and thousands of moves every year.

Position traders pay more attention to the long-term effectiveness of the asset. From this point of view, traders are closer to long-term investors than to other traders. As various types of traders, position traders also use technical analysis, which can help identify asset price trends that allow the trader to make a profit. In addition, it aims to identify trends that will last long enough, and provides

warning signals about potential changes in trends. Technical analysis usually gives traders two options: to trade assets with strong trend potential that have not yet begun to trade or to trade assets that have already begun to trade. The first option may provide higher returns, but it is riskier and requires more research.

On the other hand, the second option is less research-intensive, but the trader may miss the momentum to earn substantial profits. Similar to different trading strategies, position trading is associated with some risks [39]. The most common risks of position trading are:

- Trend reversal: An unexpected trend reversal in asset prices can result in substantial losses for the trader.
- Low liquidity: The capital of position traders is usually locked up for relatively long periods.

One of the most used position trading strategies is matching support and resistance levels. Support is the price at which the asset usually does not fall, and resistance is the point at which the price of the asset stops growing. Figure 4 visually shows the support and resistance levels. These levels help traders recognize when an asset's price movement is more likely to reverse into a downward trend or increase into an upward trend.

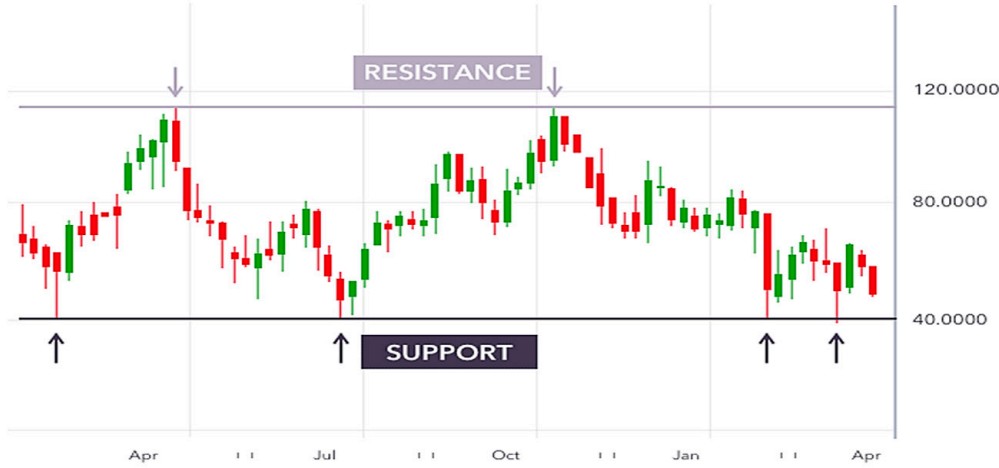

**Figure 4.** Support and resistance trading strategy.

Similar to day trading, in position trading, no exactly known factor indicates where to buy or sell. The position trader uses technical analysis for defining price movement area. Trader's decision to buy or sell the assets depends on his knowledge.

Experience shows that performing actions relied on classical strategies is not always profitable. As mentioned above, the goal of traders is to prevent their capital from "losing" and maximize incomes by using whatever system. To achieve the traders' goal, we would like to suggest a trading module built based on the deep reinforcement learning approach.

## 3. Proposed Deep Reinforcement Learning Module

This paper tells about our developed a trading agent that tries to maximize short-term profits in the cryptocurrency market. The trading agent took in features related to market conditions, the relationship of the cryptocurrency in a pair, and the agent's current financial position. Over time, the agent learned how to optimally make investment decisions—to buy, to sell, or to hold the cryptocurrency. We interactively visualized the value of our trained model portfolio. We have developed our application by using the Python programming language that gives many benefits when used machine learning concepts. The model was built with the help of TensorFlow libraries and Keras API that gives a great opportunity to the programmer for structuring deep neural network models.

In this section, we will quickly abridge what reinforcement learning and deep reinforcement learning are and show how deep reinforcement learning can be applied in a cryptocurrency market setting. Finally, we will find out the application's neural model structure.

### 3.1. Algorithm of the Proposed Model

Reinforcement learning is one of the types of machine learning and a branch of Artificial Intelligence [40]. In this type of machine learning, the machine itself learns how to behave in the environment by performing actions and comparing them with the results. It is like a machine performing trial and error strategy to determine the best action possible based on the experience. The reinforcement learning model consists of a few components: possible states of the environment S, actions A the agent can take, a reward function $R: S \times A \to R$, and a policy function $\pi$ for the agent. At each timestamp $t$, the agent takes action $a_t$, the state of the environment will change to $s_{t+1}$ from $s_t$, and the agent will receive reward $r_{t+1}$.

In a cryptocurrency market setting, the environment is all the market participants and their status. Clearly, a trading agent will have no way to know the $s_t$ since it only knows what it holds and cannot know what others hold. Possible actions in a stock market setting are buying, selling, and holding. The reward can be a value of subtracting selling price with purchasing. The ultimate goal is to maximize short-term accumulated returns $\sum_{k=0}^{inf} \gamma^k r_k$, where $\gamma$ is a discount factor (present value is preferred to equivalent future value), since the environment is partially observed, the policy function, which determines what to do, is stochastic. For example, the outputs of $\pi$ may be buy with probability 0.8, sell 0.1, and hold 0.1. We will need to sample an action according to such probability.

Neural networks are the solution to most of the complex problems in Artificial Intelligence, like computer vision, machine translation, etc. If neural networks combined with reinforcement learning, then it is straightforward to solve even more complex problems. This way of integrating neural networks with reinforcement learning is known as deep reinforcement learning [41].

Deep reinforcement learning involves the following terms:

1.  Agent: One who performs an action. In the case of games, the character of the game is an agent
2.  Action: It is the step the agent performs to achieve the reward.
3.  Rewards: These are given to the agent on reaching a particular level or performing a particular action. Rewards are considered as a measure of success.
4.  Environment: This is a world where the agent performs actions.
5.  State: It is the immediate or the present situation of the agent.
6.  Neural Network: "Brain" of the agent that can "learn" during the trade process

### 3.2. Architecture of DRL Application

The deep learning system of the trading process consists of two parts:

-   Environment—it is a class that maintains the status of investment and capital. The environment is responsible for accounting for stock assets, money management, model monitoring, buying, holding, selling stocks, and calculating the reward for actions taken.
-   Agent—it is an object that is used for communication with the environment. The agent runs every day, observes the environment, to choose an action with the policies learned during the training. The environment monitoring, determining actions with policies, recording and calculating rewards with discounted rewards, computing the gradient, and updating the network of systems with gradients—these are jobs that the agent is responsible for.

For agent communication, we consider hourly stock information as the environment. There are three types of actions for the agent: buy, hold, and sell the stocks for the interaction with the environment. The observing and monitoring of the environment would be with taking stock closing price as inputs and selecting an action by the trained agent. The working structure of the deep reinforcement learning application is described in Figure 5.

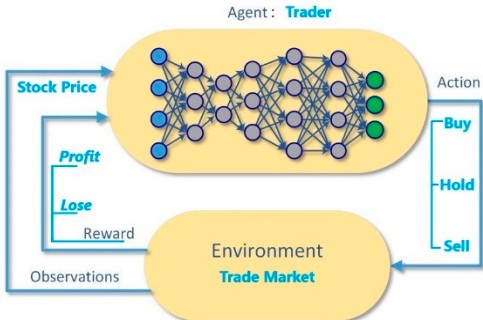

**Figure 5.** Deep reinforcement learning structure for cryptocurrency trading.

The algorithm of the application works as follows. First, an agent will get an initial state $s_0$ then will choose one of three actions randomly $a_0$. After acting, it will get a new state $s_1$ as before and performs another action $a_1$. It will repeat this act like above until states are over. However, actions are not going to be acted randomly anymore. An agent will perform an action based on rewards that are returned from the environment. Every time after occurring "buy" action, an agent will wait until the "sell" occurs. As soon as the "sell" action is performed, the agent takes selling value with the last purchased value and subtracts them. If the subtraction result is higher than zero r > 0, it means the trader gets a profit by selling. Therefore the agent will be rewarded the value of the result for the right performing action equal to the value of the result. The higher the subtracting result, the higher the agent's reward will be. Conversely, if the subtraction result is lower than zero r < 0, an agent will be punished for wrong performed action equal to the value of the result. Proceeding along these lines, an agent turns out to be progressively more "expert" than previously and shows signs of improvement results.

### 3.3. Deep Neural Model of DRL Model

The base model is a set of four multilayer models. The problem statement is built in such a way that there are three workable future price changes. Per time a signal is received, precisely one of these classes occurs, raise in price, decrease in price, or no movement.

The first layer is built with 64 hidden units and the second one with 32. The third layer involves eight neurons. The last layer, accordingly, with the number of possible actions, contains three units. As an activation function uses a Reclified Liner Unit (ReLU) function in the first three hidden layers and a linear function in the last layer. The mse—Mean Square Error function uses as an error function. The final results of all four models are used to assess confidence indicators for each of the three available outcomes. Those predictions in which the result with the highest confidence indicator are less than the threshold is considered not good enough to take action. They are marked as no-confidence predictions, and our behavior, in this case, is the same as our behavior when the prediction is no movement. Figure 6 describes the graphical structure of the proposed model.

The reward process plays a leading role in determining the effectiveness of the model. Considering on reward function, we decided the agent gets a reward for its behaviors as follows:

As stated above, every time an agent acts one of three actions: buy or sell or hold (no acts). Relatively the reward of its actions would also be different from each other by value. According to application architecture, the estimation of agent acts provides only each time after "sell" action happening by subtracting "sell" price with last "buy" price. So, to prevent the market from many

series of "buy", and increasing the number of rewards, we have included a useful additional penalty. In addition, these penalties serve to improve the performance of agents and works by counting on sequential purchases. If their number is bigger than the limit (in our case 20), the agent will receive a negative reward. The limit defined by the type of data set. For instance, the used data set that describes information per minute should be specified a bigger limit than an hourly data set.

Following the application's algorithm, an agent gets zero feedback from the environment when they chose "hold" action. Simultaneously agent "hold" action status will be under control. If an action repeats consistently several times, more precisely twenty times, an agent will be punished with a negative reward. As soon as an agent decides to "buy" coin from the market, the "hold" actions counter will be annulled.

Last but not least, each time after the agent's "sell" action, an agent gets a reward from the environment, whether negative or positive. The value of the award depends on the profitability of selling action. As mentioned above, the useful value of action is calculated by subtraction of selling price with the last purchasing price. The next job that occurs after "sell" action is annulling of "buy" and "hold" action counter.

Below given in Algorithm 1 is a pseudo-code that contains all of the information mentioned above about the reward function.

---

**Algorithm 1.** Reward Function Algorithm

---

1: ***repeat*** *(until data is over) // start trading*
2: *reward = 0*
3: *number_of_purchases = 0*
4: *number_of_sales = 0*
5: *number_of_holds = 0*
6: *choose one of three actions: // buy, sell, and hold*
7: *if (action = "buy"):*
8: number_of_purchases++
9: *// Add to the number_of_purchases one*
10: number_of_holds = 0
11: *// Declare number_of_holds as zero*
12: *if (number_of_purchases > 20):*
13: *Decrease the reward value*
14: *if (action = "hold"):*
15: number_of_holds++
16: *// Add to the number_of_holds one*
17: *if (number_of_holds > 20):*
18: *Decrease the reward value*
19: *if (action = "sell"):*
20: number_of_holds = 0
21: *// Declare number_of_holds as zero*
22: number_of_purchases = 0
23: *// Declare number_of_purchases as zero*
24: *reward ← (sell price) – (last purchase price)*
25: ***end loop***

---

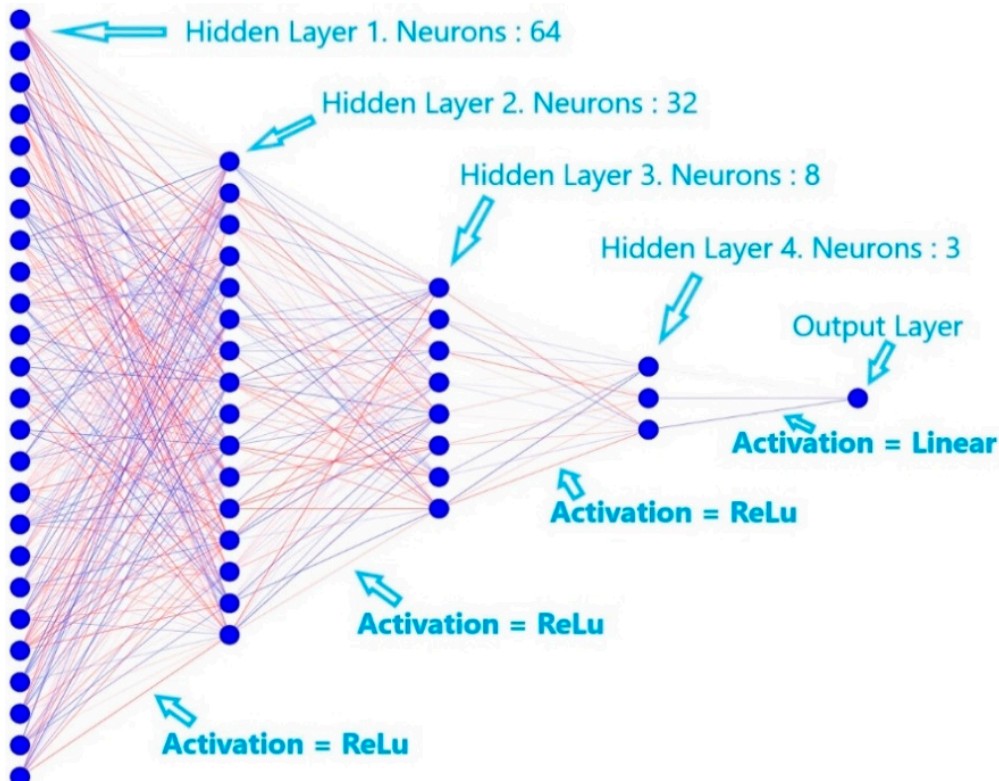

**Figure 6.** The graphical structure of the deep reinforcement learning (DRL) Application Neural Model.

## 4. Experiments and Results

To do an experiment with the deep reinforcement learning application, we downloaded three—Bitcoin, Litecoin, and Ethereum—hourly historical data from (https://www.cryptodatadownload.com). First, we divided our data into two parts. One part included much more data than another. For "teaching" an agent and allowing it to have "practice", we used the first big part of our data. The training procedure goes as follows; an agent takes the first 10-row price and does an action for each row. After that agent skips first data and appends row with 11th data and trade. Going in that order, it trades until the data is over. That is the first job that the agent has performed. An agent keeps repeating that process over 500 times.

After the training procedure, we tested the application with the second part of our data that we prepared before. As mentioned in the main idea of the algorithm, an agent must learn from experience and improve itself over time. So, we prefer to provide how results are changing over the training process.

The application creates a module file that involves the learned ability of agents in every ten times of training. Additionally, every module file keeps agents all "experience" knowledge from the beginning to the current training level. For showing results' changing, we tested our data with the different trained models. Below Figure 7a–d shows the visual information about testing with different trained models.

Finally, Figure 8 visually illustrates the best trade results that tested with the 450th training time model.

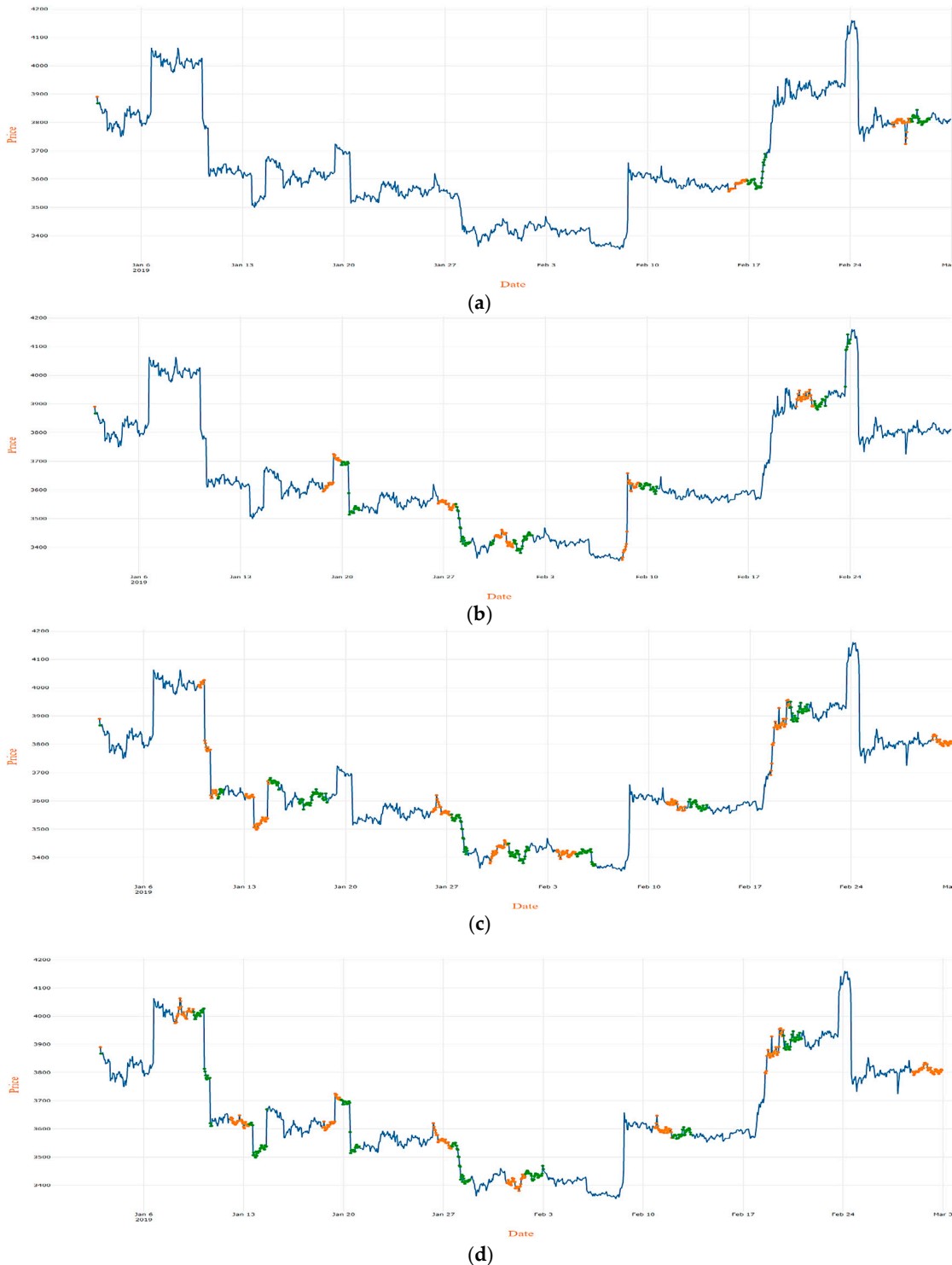

**Figure 7.** Plotting results after testing with different training time. Buy actions set with green and sell actions with yellow: (**a**) 70th training time; (**b**) 80th training time; (**c**) 90th training time; (**d**) 100th training time.

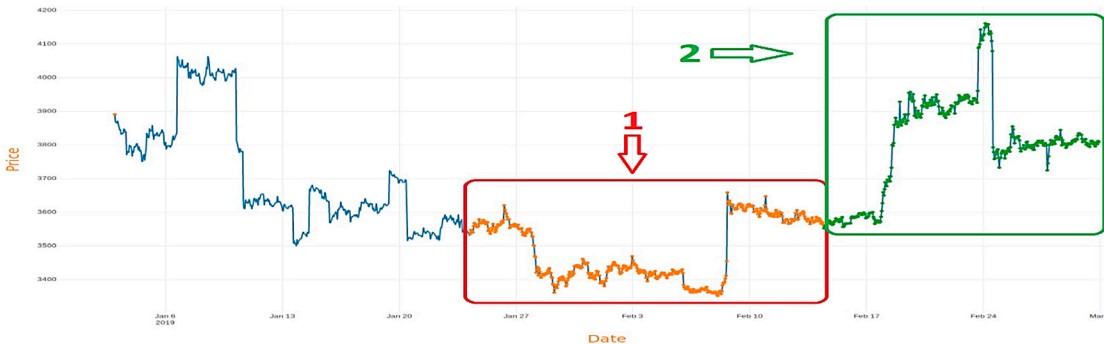

**Figure 8.** The result trading process that tested after 450-time training, where "buy" action matched the first square, "sell" in the second square.

Additionally, Table 3 shows the main trade process information with different training times. Table 4 involves the column called invested money for trading includes sum of cash with value of available coins at that moment; number of purchases and sell actions; sum of trading fees which takes whenever trader acts and contains 0.15% of trading value; money after trading which includes sum of all cash and available coin's value at the finish of trade process; quality of trade process whether invested capital grows or lose. For evaluating the trading process in Table 3, it is shown two cases with each training time. The first one shows how invested money changes when the trader's act goes based on application guidance. The second case describes when a trader holds invested money and coins until the final moment of trading without performing any actions and compares the initial investment with the final capital.

**Table 3.** Main information about the trading process tested with different training time.

| Training Time | Invested Money for Trading ($) | Trade with | "Buy" Actions | "Sell" Actions | Transaction Fee ($) | Money after Trading ($) | Quality of Trading (%) |
|---|---|---|---|---|---|---|---|
| 70-time | 1 000 000 | DC strategy | 63 | 63 | 699.2 | 1 000 266 | 100.02 (Grew 0.02) |
| | | hold position | 0 | 0 | 0 | 980,637 | 98 (Lost 2) |
| 80-time | 1 000 000 | DC strategy | 150 | 150 | 1 626.9 | 996 214.8 | Lost 0.4 |
| | | hold position | 0 | 0 | 0 | 980 637 | Lost 2 |
| 90-time | 1 000 000 | DC strategy | 262 | 230 | 2 663.1 | 996 029.8 | Lost 0.4 |
| | | hold position | 0 | 0 | 0 | 980 637 | Lost 2 |
| 100-time | 1 000 000 | DC strategy | 268 | 218 | 2 677.9 | 994 529.5 | Lost 0.6 |
| | | hold position | 0 | 0 | 0 | 980 637 | Lost 2 |
| 450-time | 1 000 000 | DC strategy | 510 | 396 | 5 577.0 | 1 155 658 | 115.5 (Grew 15.5) |
| | | hold position | 0 | 0 | 0 | 980 637 | 98 (Lost 2) |

**Table 4.** Experiment results on Bitcoin's data.

| | Invested Money ($) | Number of Actions | Money after Trading ($) | Quality of Trading % |
|---|---|---|---|---|
| Double Cross Strategy | 48 124 | 14 | 48 563 | 100.9 (Grew 0.9) |
| Swing Trading | 48 124 | 5 | 48 469 | 100.7 (Grew 0.7) |
| Scalping Trading | 48 124 | 130 | 51 073 | 106.1 (Grew 6.1) |
| DRL Application | 1 000 000 | 400 | 1 144 961 | 114.4 (Grew 14.4) |

For making sure and defining how useful using deep reinforcement learning approach, we took the same period and tested simultaneously with three classical strategies: the double cross strategy, swing trading, and scalping trading.

We tested on Bitcoin with the period from the 19 of March to the 21 of April and with the 450th time training model. Table 4 shows 14 actions brought 0.9% profit to the trader when the double cross strategy was used. At that time, the swing trading showed five actions and ended up with a 0.7% profit. The most satisfactory result between classical approaches was in the scalping trading. During the experiment occurred 130 actions, which brought 6.1% of profit for the trader. In the same period, the deep reinforcement learning (DRL) application performed 400 operations. As a result, the trader's capital grew by 14.4%.

Then we extended our experiment by examining Litecoin and Ethereum. As in Bitcoin's case, we trained agents separately with their separate historical data and tested for Litecoin with the period between the 19 of March and 21 of April as well. The results are described in Table 5. By using the double cross strategy, the trader completed 12 actions and lost 2.2% of the money invested within a month. In the same period, the swing trader made four trade and finished with a 3.6% profit. The experiment with scalping trading strategy ended with 23.7% profit and 134 trades. During that period, the DRL application performed 642 actions. As a result, the trader earned 74.6% more money than the invested money.

**Table 5.** Experiment results on Litecoin's data.

| | Invested Money ($) | Number of Actions | Money after Trading ($) | Quality of Trading % |
|---|---|---|---|---|
| Double Cross Strategy | 10 814 | 12 | 10 576 | 97.8 (Lost 2.2) |
| Swing Trading | 10 814 | 4 | 11 217 | 103.6 (Grew 3.6) |
| Scalping Trading | 10 814 | 134 | 13 382 | 123.7 (Grew 23.7) |
| DRL Application | 10 000 | 642 | 17 467 | 174.6 (Grew 74.6) |

For Ethereum, we chose a time from the 19 of March to the 21 of April. The experiment results are shown in Table 6. Ethereum's case also changed in a good way. During the trade based on the double cross strategy, there are nine cross points occurred and finished with a 2.9% decrease for initial money. For that period, a swing trader did four actions and lost 9.4% of the capital. The scalpers' 112 actions also finished with lost 2.4% of their investments. In consequence of the DRL application, the trader acted 294 times and got 41.4% profit.

**Table 6.** Experiment results on Ethereum's data.

| | Invested Money ($) | Number of Actions | Money after Trading ($) | Quality of Trading % |
|---|---|---|---|---|
| Double Cross Strategy | 11 739 | 9 | 11 400 | 97.1 (Lost 2.9) |
| Swing Trading | 11 739 | 4 | 10 643 | 90.6 (Lost 9.4) |
| Scalping Trading | 11 739 | 112 | 11 462 | 97.6 (Lost 2.4) |
| DRL Application | 10 000 | 294 | 14 140 | 141.4 (Grew 41.4) |

## 5. Conclusions

This article proposes an application works based on a deep reinforcement learning algorithm for the cryptocurrency market, which aims to make the right decision during the trade process. The goal is achieved and verified through experiments.

According to the results, a trader's financial status has got better after using deep reinforcement learning applications. Experiment result on Bitcoin shows an application detected 400 cases useful for purchase and sale and gave a chance to earn additionally 144 thousand dollars or 114.4 % more money than invested money within one month. The performance of Bitcoin and the agent's decision during the experiment is shown in Figure 9.

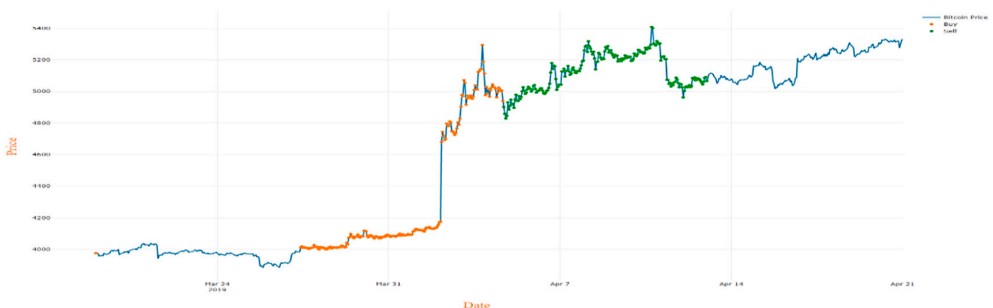

**Figure 9.** Experiment results on Bitcoin. Yellow point present purchases, green points present "sell".

During a test on Litecoin, an application found 642 cases right for trade, which is cause to increase initial fortune by 174.6%. Figure 10 illustrates the sequence of the agent's actions and the performance of Litecoin.

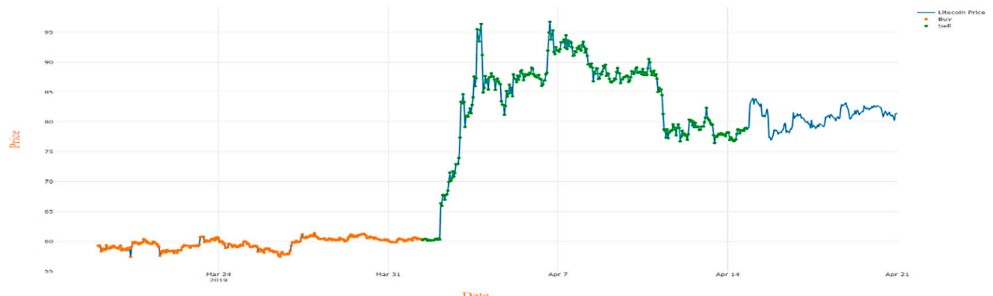

**Figure 10.** Experiment results on Litecoin. Yellow point present purchases, green point presents "sell".

Finally, the experiment result on Ethereum indicates that within a month, there were 294 useful trades that happened, and they lead to a finish with 41.4% net returns. Graphic of the experiment is shown in Figure 11.

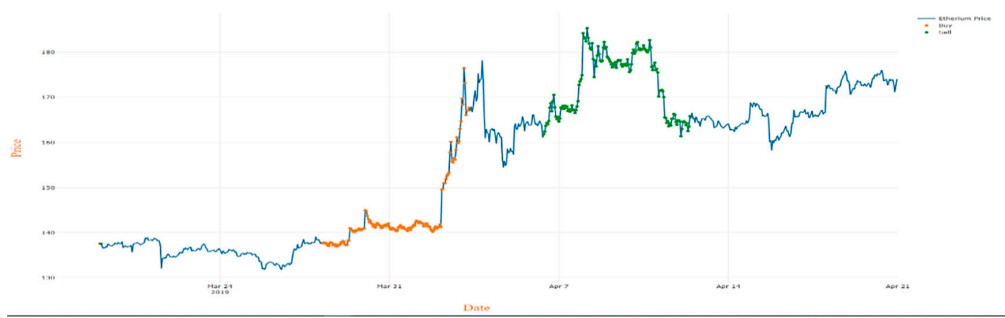

**Figure 11.** Experiment results on Ethereum. Yellow point present purchases, green point present "sell".

In summary, our proposed model can serve to help the trader choose correctly one of three actions: buying, selling, and holding the stock. In other words, a trader gets a recommendation about the right option and can catch up the good points for trading.

However, the graph of the trade process tells us that instead of some "hold" acting time, "buy" or "sell" options will be useful for increasing the trader's investment. So, in the future, we need to improve our algorithm to maximize the trader's profit.

**Author Contributions:** Writing—original draft preparation, software, O.S.; methodology, resources, data curation, O.S., C.W.L., H.K.K., and A.M.; conceptualization, formal analysis, R.O., J.A. and H.S.J.; investigation, writing—review and editing, H.J.O. and H.S.J. All authors have read and agreed to the published version of the manuscript.

**Funding:** This research was supported by Basic Science Research Program through the National Research Foundation of Korea (NRF) funded by the Ministry of Education (2018R1D1A1B07043417).

**Conflicts of Interest:** The authors declare no conflict of interest.

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
