# Peer review of "Recommending Cryptocurrency Trading Points with Deep Reinforcement Learning Approach"

_applsci, doi:10.3390/app10041506_

Round 1

Reviewer 1 Report

No major concerns after this revision.

Author Response

Thank you for your review. We made a few changes on grammatical errors and added figures that allows to better understand of experiment process. All changes are highlighted.

Reviewer 2 Report

The explanation of the theoretical background can be improved in terms of better clarify the diversity or similarity between the different approach in literature.
Also, the analysis results can be improved: I suggest to compare data in the same timespan for Bitcoin, Litecoin and Ethereum and to show the performance of this cryptocurrencies.

Author Response

The explanation of the theoretical background can be improved in terms of better clarify the diversity or similarity between the different approach in literature.

: [answer] Thank you for your review. We added extra information about the background of our research, as you mentioned. All changes are highlighted.

Also, the analysis results can be improved: I suggest to compare data in the same timespan for Bitcoin, Litecoin and Ethereum and to show the performance of this cryptocurrencies.

: [answer] We took the same time period for the experiment and applied for all types of cryptocurrencies (Bitcoin, Litecoin, and Ethereum). The additional graphs that show the experiment process were included for showing the performance.

This manuscript is a resubmission of an earlier submission. The following is a list of the peer review reports and author responses from that submission.

Round 1

Reviewer 1 Report

I can see that the language has improved and I understand the manuscript better. Now my primary concern is the methodology although the language still  needs to be improved as well.

You are comparing a deep learning reinforcement algorithm with a strategy borrowed from technical analysis which may not be an appropriate benchmark.

First of all, you do not have any academic reference for your benchmark, the double crossover strategy, which seems to be solely based on  industrial  expert's knowledge. Secondly, although the double crossover strategy is common in practice, you are ignoring other common strategies.

Reviewer 2 Report

See attached letter
